# A genome-wide association study identifies six novel risk loci for primary biliary cholangitis

Fang Qiu[1,*], Ruqi Tang[2,*], Xianbo Zuo[3,*], Xingjuan Shi[1,*], Yiran Wei[2,*], Xiaodong Zheng[3,*], Yaping Dai[4], Yuhua Gong[5], Lan Wang[6], Ping Xu[7], Xiang Zhu[7], Jian Wu[8], Chongxu Han[9], Yueqiu Gao[10], Kui Zhang[11], Yuzhang Jiang[12], Jianbo Zhou[13], Youlin Shao[14], Zhigang Hu[15], Ye Tian[16], Haiyan Zhang[2], Na Dai[17], Lei Liu[18], Xudong Wu[19], Weifeng Zhao[8], Xiaomin Zhang[20], Zhidong Zang[21], Jinshan Nie[22], Weihao Sun[23], Yi Zhao[24], Yuan Mao[25], Po Jiang[26], Hualiang Ji[27], Qing Dong[28], Junming Li[29], Zhenzhong Li[30], Xinli Bai[30], Li Li[31], Maosong Lin[32], Ming Dong[1], Jinxin Li[1], Ping Zhu[1], Chan Wang[1], Yanqiu Zhang[1], Peng Jiang[1], Yujue Wang[1], Rohil Jawed[1], Jing Xu[1], Yu Zhang[1], Qixia Wang[2], Yue Yang[2], Fan Yang[2], Min Lian[2], Xiang Jiang[2], Xiao Xiao[2], Yanmei Li[2], Jingyuan Fang[2], Dekai Qiu[2], Zhen Zhu[14], Hong Qiu[6], Jianqiong Zhang[1], Wenyan Tian[8], Sufang Chen[7], Ling Jiang[8], Bing Ji[6], Ping Li[6], Guochang Chen[18], Tianxue Wu[9], Yan Sun[9], Jianjiang Yu[13], Huijun Tang[13], Michun He[8], Min Xia[15], Hao Pei[4], Lihua Huang[4], Zhuye Qing[25], Jianfang Wu[33], Qinghai Huang[1], Junhai Han[1], Wei Xie[1], Zhongsheng Sun[34], Jian Guo[35], Gengsheng He[36], M. Eric Gershwin[37], Zhexiong Lian[38], Xiang Liu[39], Michael F. Seldin[37], Xiangdong Liu[1], Weichang Chen[8] & Xiong Ma[2]

Primary biliary cholangitis (PBC) is an autoimmune liver disease with a strong hereditary component. Here, we report a genome-wide association study that included 1,122 PBC cases and 4,036 controls of Han Chinese descent, with subsequent replication in a separate cohort of 907 PBC cases and 2,127 controls. Our results show genome-wide association of 14 PBC risk loci including previously identified 6p21 (HLA-DRA and DPB1), 17q12 (ORMDL3), 3q13.33 (CD80), 2q32.3 (STAT1/STAT4), 3q25.33 (IL12A), 4q24 (NF-κB) and 22q13.1 (RPL3/SYNGR1). We also identified variants in IL21, IL21R, CD28/CTLA4/ICOS, CD58, ARID3A and IL16 as novel PBC risk loci. These new findings and histochemical studies showing enhanced expression of IL21 and IL21R in PBC livers (particularly in the hepatic portal tracks) support a disease mechanism in which the deregulation of the IL21 signalling pathway, in addition to CD4 T-cell activation and T-cell co-stimulation are critical components in the development of PBC.

[1] Key Laboratory of Developmental Genes and Human Diseases, Institute of Life Sciences, Southeast University, Nanjing, Jiangsu 210096, China. [2] Department of Gastroenterology and Hepatology, Renji Hospital, School of Medicine, Shanghai Jiao Tong University, Shanghai Institute of Digestive Disease, Shanghai 200001, China. [3] Department of Dermatology at No. 1 Hospital, Institute of Dermatology, Anhui Medical University, Hefei, Anhui 230022, China. [4] Department of Laboratory Medicine, The Fifth People's Hospital of Wuxi, Wuxi, Jiangsu 214005, China. [5] Department of Laboratory Medicine, The Third People's Hospital of Zhenjiang, Zhenjiang, Jiangsu 212005, China. [6] Department of Laboratory Medicine, The 81th Hospital of PLA, Nanjing, Jiangsu 210002, China. [7] Department of Laboratory Medicine, The Fifth People's Hospital of Suzhou, Soochow University, Suzhou, Jiangsu 215007, China. [8] Department of Rheumatology, Department of Gastroenterology, The First Affiliated Hospital of Soochow University, Suzhou, Jiangsu 215006, China. [9] Department of Laboratory Medicine, Subei People's Hospital, Clinical Medical College, Yangzhou University, Yangzhou, Jiangsu 225001, China. [10] Department of Hepatology, Shuguang Hospital, Shanghai University of Traditional Chinese Medicine, Shanghai 200021, China. [11] Department of Laboratory Medicine, Nanjing Drum Tower Hospital, The Affiliated Hospital of Nanjing University Medical School, Nanjing, Jiangsu 210008, China. [12] Department of Laboratory Medicine, Huai'an First People's Hospital, Nanjing Medical University, Huai'an, Jiangsu 223300, China. [13] Department of Laboratory Medicine, Jiangyin People's Hospital, Southeast University, Jiangyin, Jiangsu 214400, China. [14] Department of Laboratory Medicine, The Third People's Hospital of Changzhou, Changzhou, Jiangsu 213000, China. [15] Department of Laboratory Medicine, Affiliated Wuxi People's Hospital of Nanjing Medical University, Wuxi, Jiangsu 214023, China. [16] Department of Radiology, The Second Affiliated Hospital of Soochow University, Suzhou, Jiangsu 215004, China. [17] Department of Gastroenterology, Jiangsu University affiliated Kunshan Hospital, Kunshan, Jiangsu 215300, China. [18] Department of Gastroenterology, Yixing People's Hospital, Yixing, Jiangsu 214200, China. [19] Department of Gastroenterology, Yancheng First People's Hospital, Yancheng, Jiangsu 224005, China. [20] Department of Laboratory Medicine, The University Hospital, Southeast University, Nanjing, Jiangsu 210096, China. [21] Department of Hepatology, The Second Hospital of Nanjing, Southeast University, Nanjing, Jiangsu 210003, China. [22] Department of Gastroenterology, Taicang First People's Hospital, Soochow University, Taicang, Jiangsu 215400, China. [23] Department of Gastroenterology, The First Affiliated Hospital of Nanjing Medical University, Nanjing, Jiangsu 210029, China. [24] Department of Gastroenterology, Eastern Hepatobiliary Surgery Hospital, Shanghai 201805, China. [25] Department of Immunology, Nanjing Kingmed Clinical Laboratory Co. Ltd. Nanjing, Jiangsu 210042, China. [26] Department of Hepatology, The Second People's Hospital of Jingjiang, Jingjiang, Jiangsu 214500, China. [27] Department of Gastroenterology, Hai'an People's Hospital, Nantong University Medical School, Hai'an, Jiangsu 226600, China. [28] Department of Laboratory Medicine, Suzhou Hospital of Traditional Chinese Medicine, Suzhou, Jiangsu 215009, China. [29] Department of Laboratory Medicine, The First Affiliated Hospital of Nanchang University, Nanchang, Jiangxi 330006, China. [30] Department of Paediatrics, The Second Hospital of Hebei Medical University, Shijiazhuang, Hebei 050000, China. [31] Department of Laboratory Medicine, Zhongda Hospital, School of Medicine, Southeast University, Nanjing, Jiangsu 210009, China. [32] Department of Gastroenterology, Taizhou People's Hospital, Taizhou, Jiangsu 225300, China. [33] Department of Hepatology, Traditional Chinese Medicine Hospital of Kunshan, Kunshan 215300, China. [34] Department of Genomics and Epigenomics, Beijing Institutes of Life Science, Chinese Academy of Sciences, Beijing 100101, China. [35] Department of Gerontology, Beijing Hospital, Beijing 100730, China. [36] Department of Nutrition and Health, School of Public Health, Fudan University, Shanghai 200032, China. [37] Division of Rheumatology, Allergy, and Clinical Immunology, Rowe Program in Genetics, University of California-Davis, Davis, California 95616, USA. [38] Department of Immunology, School of Life Sciences, University of Science and Technology of China, Hefei 230027, Anhui, China. [39] Department of Stomatology, The First Affiliated Hospital, Hainan Medical University, Haikou, Hainan 571199, China. * These authors contributed equally to this work. Correspondence and requests for materials should be addressed to X.L. (email: xiangdongliu@seu.edu.cn) or to W.C. (email: weichangchen@126.com) or to X.M. (email: maxiongmd@163.com).

Primary biliary cholangitis (PBC) is an autoimmune disease characterized by chronic inflammation and destruction of intrahepatic bile ducts. It is often considered a model autoimmune disease because of its specific autoantibodies (the anti-mitochondrial antibodies) and distinctive bile duct pathology. PBC shows a strong genetic predisposition, with the highest concordance rate of all autoimmune diseases among identical twins[1]. Several genome-wide association studies (GWAS) and Immunochip studies have revealed a nearly uniform level of genetic susceptibility among people (or populations) of different European ancestries[2–7], showing significant association of more than 20 loci with PBC. However, a PBC GWAS in Japanese (including 487 PBC cases and 476 controls and replication in 787 cases and 615 controls) showed very different results identifying only TNFSF15 and POU2AF1 as the main susceptibility loci, neither of which was described in the European studies[8]. Recently, we performed a replication study of the 14 most significant single nucleotide polymorphisms (SNPs) identified in either or both of the European and Japanese cohorts, using 1,070 Han Chinese PBC cases and 1,198 healthy controls, and confirmed strong association of the TNFSF15, CD80 and 17q12 loci with the disease in Chinese patients, with P-value reaching GWA significance ($P < 5 \times 10^{-8}$) at TNFSF15 and CD80 loci[9]. These early results also ensured required samples for our subsequent GWAS design.

To identify additional genetic risk factors for PBC and further define differences in susceptibility to the disease between European and East Asian populations, we conducted a large case–control GWAS of 1,127 PBC subjects and 4,074 controls of self-declared Han Chinese origin with population-specific HumanOmniZhongHua-8 beadchip and verified the results by genotyping top SNPs in a separate cohort of 907 PBC cases and 2,127 controls. We confirmed eight previously identified PBC susceptibility loci and also identified variants in IL21, IL21R, CD28/CTLA4/ICOS, CD58, ARID3A and IL16 as novel PBC risk loci. Subsequent histochemical studies show enhanced expression of IL21 and IL21R in PBC livers. These findings significantly expand our understanding of the disease susceptibility and suggest IL21 signalling pathway, in addition to CD4 T-cell activation and T-cell co-stimulation, are critical components in the development of PBC.

## Results

**Genome-wide discovery analysis.** The HumanOmniZhongHua-8 beadchip (v1.1) contains 894,956 markers covering common, intermediate and rare variations found within Chinese populations. Following stringent quality control measures, 776,516 autosomal SNPs were available across 1,122 cases and 4,036 geographically matched controls (Supplementary Tables 1–3; Supplementary Fig. 1). Principal-component analysis (PCA) did not demonstrate substantial stratification in our study population ($\lambda_{gc} = 1.029$, and see Supplementary Fig. 2). The quantile–quantile plot of the case–control test showed a substantial excess of significant associations in the tail of the distribution even after removal of the major histocompatibility complex (MHC) region that includes human leukocyte antigen (HLA) genes (Supplementary Fig. 3).

We identified 179 SNPs in the MHC region and 39 SNPs in 8 non-MHC loci exceeding genome-wide significance in the discovery stage (Fig. 1; Supplementary Tables 4–7). The most significant association was found in the HLA-DRA locus. A SNP within HLA-DRPB1 was the second most significant locus in the MHC region. Two SNPs from the MHC region and 32 SNPs from 24 non-MHC loci with at least suggestive association ($P < 5 \times 10^{-6}$) were selected for replication studies (Supplementary Table 6). There were 907 PBC cases and 2,127 controls in the replication panels that confirmed association of the MHC region and 15 non-MHC loci. Each of these loci in the combined analysis (2,029 PBC cases and 6,163 controls) also met genome-wide significance. These included 8 non-MHC risk loci previously identified in Caucasian and Japanese cohorts (Supplementary Fig. 4)[2–8]. In addition, a previously identified risk locus, DDX6-CXCR5, did not replicate but still achieved genome-wide significance ($P = 2.56 \times 10^{-13}$) in the combined analysis (Table 1; Supplementary Fig. 4).

Among the non-MHC risk loci, the strongest association was observed in rs4979467, a SNP in the TNFSF15-TNFSF8 locus, the same as reported in the Japanese cohort[8]. This locus has been implicated in several autoimmune diseases[10]. Two previous studies have reported that two functional SNPs, rs6478108 and rs4979462, in the TNFSF15 promoter region are associated with TNFSF15 expression[11,12]. The rs4979467 SNP is in strong linkage disequilibrium (LD) with rs6478108 ($r^2 = 1$) and rs4979462 ($r^2 = 0.68$) in the Han Chinese population, implying that the

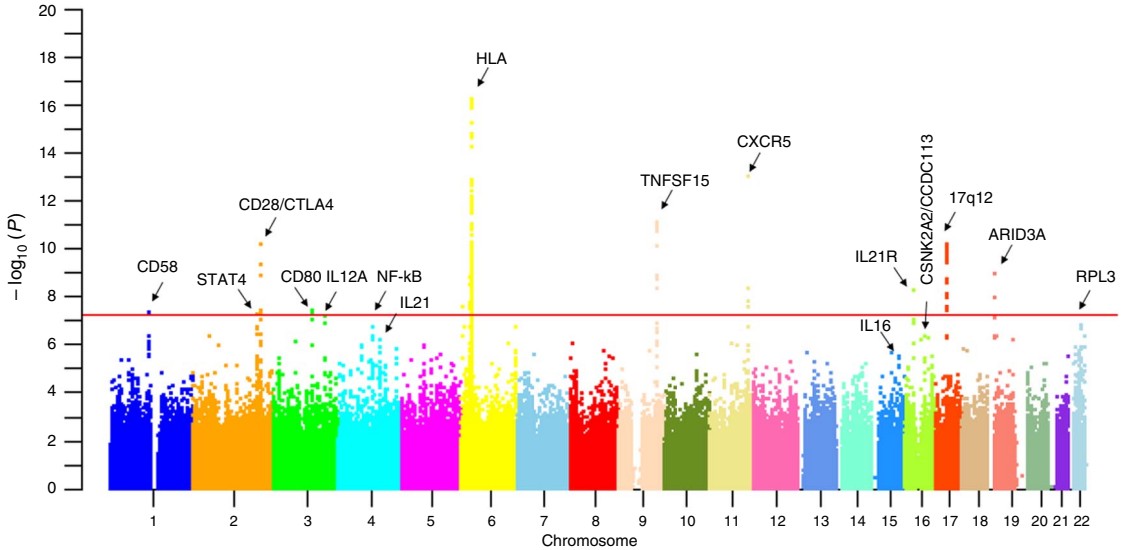

**Figure 1 | Manhattan plot of PBC GWAS data.** Genome-wide association results from the initial GWA analysis. The genome-wide P-values of the Cochran-Armitage trend test from 796,160 polymorphic SNPs in 1,122 PBC cases and 4,036 controls of Han Chinese ancestry are presented. The chromosomal distribution of all the P-values (− log10 P-values) is shown.

**Table 1 | Known PBC loci reaching genome-wide significance.**

| Chr | SNP | MA | Position* (bp) | Genes in the region | GWAS cohort | | | | Replication cohort | | | | Combined samples | |
|---|---|---|---|---|---|---|---|---|---|---|---|---|---|---|
| | | | | | Case MAF | Control MAF | P-value | OR (95% CI) | Case MAF | Control MAF | P-value | OR (95% CI) | P-value | OR (95% CI) |
| 6p21 | rs9268644 | A | 32408044 | HLA-DRA | 0.098 | 0.177 | $2.41 \times 10^{-19}$ | 0.51 (0.44–0.59) | 0.094 | 0.169 | $5.84 \times 10^{-14}$ | 0.51 (0.43–0.61) | $7.83 \times 10^{-31}$ | 0.51 (0.45–0.57) |
| 6p21 | rs9501251 | G | 33049663 | HLA-DPB1 | 0.082 | 0.044 | $8.17 \times 10^{-13}$ | 1.94 (1.61–2.33) | 0.091 | 0.045 | $6.18 \times 10^{-12}$ | 2.11 (1.70–2.62) | $2.10 \times 10^{-22}$ | 2.01 (1.76–2.32) |
| 9q32 | rs4979467 | A | 117630043 | TNFSF15,TNFSF8 | 0.399 | 0.322 | $8.28 \times 10^{-12}$ | 1.40 (1.27–1.54) | 0.427 | 0.230 | $1.87 \times 10^{-21}$ | 1.74 (1.55–1.95) | $1.22 \times 10^{-29}$ | 1.53 (1.42–1.64) |
| 17q12 | rs9635726 | G | 38020141 | Multiple genes | 0.382 | 0.309 | $6.42 \times 10^{-11}$ | 1.38 (1.25–1.52) | 0.376 | 0.309 | $4.24 \times 10^{-7}$ | 1.35 (1.20–1.52) | $1.62 \times 10^{-16}$ | 1.37 (1.27–1.48) |
| 11q23.3 | rs77871618 | A | 118733624 | DDX6, CXCR5 | 0.225 | 0.158 | $9.12 \times 10^{-14}$ | 1.55 (1.38–1.74) | 0.202 | 0.175 | $1.20 \times 10^{-2}$ | 1.20 (1.04–1.38) | $2.56 \times 10^{-13}$ | 1.40 (1.28–1.53) |
| 3q13.33 | rs3732421 | G | 119150089 | CD80 | 0.271 | 0.332 | $3.76 \times 10^{-8}$ | 0.75 (0.67–0.83) | 0.261 | 0.324 | $1.51 \times 10^{-6}$ | 0.74 (0.65–0.83) | $3.10 \times 10^{-13}$ | 0.74 (0.68–0.80) |
| 2q32.3 | rs10168266 | A | 191935804 | STAT1,STAT4 | 0.397 | 0.336 | $5.82 \times 10^{-8}$ | 1.31 (1.19–1.44) | 0.393 | 0.328 | $1.26 \times 10^{-6}$ | 1.33 (1.18–1.49) | $3.95 \times 10^{-13}$ | 1.31 (1.22–1.41) |
| 3q25.33 | rs582537 | C | 159710098 | IL12A | 0.234 | 0.291 | $6.55 \times 10^{-8}$ | 0.74 (0.66–0.83) | 0.237 | 0.287 | $6.83 \times 10^{-5}$ | 0.77 (0.68–0.88) | $2.36 \times 10^{-11}$ | 0.75 (0.69–0.82) |
| 4q24 | rs1598856 | G | 103446115 | NF-kB1 | 0.536 | 0.474 | $1.83 \times 10^{-7}$ | 1.28 (1.17–1.41) | 0.527 | 0.474 | $2.07 \times 10^{-4}$ | 1.23 (1.10–1.39) | $1.80 \times 10^{-10}$ | 1.26 (1.17–1.35) |
| 12p13.31 | rs4149576 | A | 6449115 | TNFRSF1A | 0.148 | 0.113 | $1.11 \times 10^{-5}$ | 1.35 (1.18–1.55) | 0.147 | 0.110 | $7.27 \times 10^{-5}$ | 1.39 (1.18–1.63) | $3.81 \times 10^{-9}$ | 1.37 (1.23–1.52) |
| 22q13.1 | rs137603 | C | 39694225 | RPL3,SYNGR1 | 0.109 | 0.152 | $2.06 \times 10^{-7}$ | 0.68 (0.59–0.79) | 0.119 | 0.144 | $1.10 \times 10^{-2}$ | 0.81 (0.68–0.95) | $2.67 \times 10^{-8}$ | 0.73 (0.65–0.81) |

Chr, chromosome; MA, minor allele; MAF, minor allele frequency.
P-value is based on additive model; OR, odds ratio, is calculated for minor allele.
*Sequence position was annotated based on the GRCh37/hg19 assembly.

**Table 2 | Novel PBC loci reaching genome-wide significance.**

| Chr | SNP | MA | Position* (bp) | Genes in the region | GWAS cohort | | | | Replication cohort | | | | Combined samples | |
|---|---|---|---|---|---|---|---|---|---|---|---|---|---|---|
| | | | | | Case MAF | Control MAF | P-value | OR (95% CI) | Case MAF | Control MAF | P-value | OR (95% CI) | P-value | OR (95% CI) |
| 16p12.1 | rs2189521 | G | 27413566 | IL4R, IL21R | 0.242 | 0.306 | $5.40 \times 10^{-9}$ | 0.73 (0.65–0.81) | 0.246 | 0.320 | $9.87 \times 10^{-9}$ | 0.69 (0.61–0.79) | $4.00 \times 10^{-16}$ | 0.71 (0.66–0.78) |
| 16p12.1 | rs10852316 | A | 27398555 | IL4R, IL21R | 0.379 | 0.442 | $8.83 \times 10^{-8}$ | 0.77 (0.70–0.85) | 0.374 | 0.444 | $5.59 \times 10^{-7}$ | 0.75 (0.67–0.84) | $2.76 \times 10^{-13}$ | 0.76 (0.71–0.82) |
| 2q33.2 | rs4675369 | A | 204643194 | CD28, CTLA4, ICOS | 0.526 | 0.448 | $6.56 \times 10^{-11}$ | 1.37 (1.24–1.50) | 0.505 | 0.452 | $1.65 \times 10^{-4}$ | 1.24 (1.11–1.38) | $1.38 \times 10^{-13}$ | 1.31 (1.22–1.41) |
| 2q33.2 | rs7599230 | G | 204648661 | CD28, CTLA4, ICOS | 0.471 | 0.408 | $9.08 \times 10^{-8}$ | 1.29 (1.18–1.42) | 0.459 | 0.411 | $5.96 \times 10^{-4}$ | 1.22 (1.09–1.36) | $3.30 \times 10^{-10}$ | 1.26 (1.18–1.36) |
| 4q27 | rs925550 | A | 123588526 | IL21 | 0.428 | 0.370 | $6.21 \times 10^{-7}$ | 1.27 (1.16–1.40) | 0.427 | 0.353 | $7.44 \times 10^{-8}$ | 1.37 (1.22–1.53) | $3.87 \times 10^{-13}$ | 1.31 (1.21–1.40) |
| 4q27 | rs17005934 | G | 123549699 | IL21 | 0.408 | 0.352 | $1.42 \times 10^{-6}$ | 1.27 (1.15–1.39) | 0.428 | 0.362 | $1.25 \times 10^{-6}$ | 1.32 (1.18–1.48) | $1.06 \times 10^{-11}$ | 1.29 (1.21–1.39) |
| 1p13.1 | rs2300747 | A | 117104215 | CD58 | 0.481 | 0.416 | $4.54 \times 10^{-8}$ | 1.30 (1.18–1.43) | 0.475 | 0.412 | $8.02 \times 10^{-6}$ | 1.29 (1.15–1.44) | $1.84 \times 10^{-12}$ | 1.29 (1.20–1.39) |
| 1p13.1 | rs10924106 | A | 117053745 | CD58 | 0.509 | 0.449 | $4.58 \times 10^{-7}$ | 1.27 (1.16–1.40) | 0.508 | 0.446 | $1.10 \times 10^{-5}$ | 1.28 (1.15–1.43) | $2.40 \times 10^{-11}$ | 1.28 (1.19–1.37) |
| 19p13.3 | rs10415976 | G | 941603 | ARID3A | 0.414 | 0.486 | $1.06 \times 10^{-9}$ | 0.75 (0.68–0.82) | 0.439 | 0.489 | $3.51 \times 10^{-4}$ | 0.82 (0.73–0.91) | $3.61 \times 10^{-12}$ | 0.77 (0.72–0.84) |
| 19p13.3 | rs10414193 | G | 939697 | ARID3A | 0.403 | 0.471 | $1.09 \times 10^{-8}$ | 0.76 (0.69–0.83) | 0.426 | 0.474 | $6.84 \times 10^{-4}$ | 0.82 (0.74–0.92) | $5.80 \times 10^{-11}$ | 0.79 (0.73–0.85) |
| 15q25.1 | rs11556218 | C | 81598269 | IL16 | 0.233 | 0.189 | $3.03 \times 10^{-6}$ | 1.31 (1.17–1.46) | 0.226 | 0.188 | $6.94 \times 10^{-4}$ | 1.26 (1.10–1.45) | $8.99 \times 10^{-9}$ | 1.29 (1.18–1.41) |
| 16q21 | rs2550374 | A | 58254448 | CSNK2A2, CCDC113 | 0.424 | 0.484 | $4.19 \times 10^{-7}$ | 0.78 (0.71–0.86) | 0.446 | 0.486 | $5.26 \times 10^{-3}$ | 0.85 (0.76–0.95) | $1.51 \times 10^{-8}$ | 0.81 (0.76–0.87) |

Chr, chromosome; MA, minor allele; MAF, minor allele frequency.
P-value is based on additive model; OR, odds ratio, is calculated for minor allele.
*Sequence position was annotated based on the GRCh37/hg19 assembly.

functional variation in the *TNFSF15* promoter can affect the genetic susceptibility to PBC.

*IL12A* was found to be strongly associated with Han Chinese PBC cohorts, in contrast to both the Japanese GWAS and our previous candidate gene study[8,9]. On the basis of the dense fine-mapping study with the Immunochip, four independent associations were observed at the *IL12A-SCHIP1* locus in European ancestry PBC cases[6]. The significant association (rs582537) in our GWAS corresponded to the third signal rs668998 in the European studies (Supplementary Table 8). This association also explained the discrepancy with the previous candidate gene study, in which selected SNPs represented only the first signal at the *IL12A* locus.

**Discovery and validation of six novel PBC risk loci.** Six novel risk loci associated with PBC were identified and validated in the current study (Table 2; Fig. 2). Both *IL21R* and *IL21* were strongly associated in the Han Chinese PBC cohorts. Notably, several recent studies have found increased production of IL21 from T follicular helper (Tfh) cells in PBC patients, which might mediate B-cell maturation and autoantibody secretion[13,14]. Multiple SNPs

in the *IL21* and *IL21R* loci were significantly associated with PBC (Supplementary Table 5). We identified two independent signals represented by variants rs925550 and rs17005934 in the *IL21* locus using stepwise conditional regression analysis with the two SNPs (Supplementary Table 9). These two signals were separated in the Han Chinese population as a result of a major recombination event (Fig. 2c). A much weaker association (rs108507092) observed in the nearby *IL2* region was presumably due to linkage disequilibrium ($r^2 = 0.64$) with rs17005934.

The most significant SNP, rs2189521, at the *IL21R* locus is located in the 5′ UTR of *IL21R*, indicating its potential involvement in regulating differential IL21R expression. *IL4R* is located 27 kb upstream from *IL21R*. To analyse if the significance came from the contribution of *IL4R*, we performed LD analysis with the Han Chinese 1000 genome sequencing data. The results indicate that *IL21R* and *IL4R* are in two separate LD blocks in the Han population (Fig. 2a). None of the significant SNPs observed at the *IL21R* locus are linked to *IL4R*, excluding the association of *IL4R* with Han PBC cohorts. The variant rs2189521 is an expression quantitative trait locus (eQTL), in strong LD ($r^2 = 0.98$–1) with multiple SNPs in the *IL21R* region, suggesting

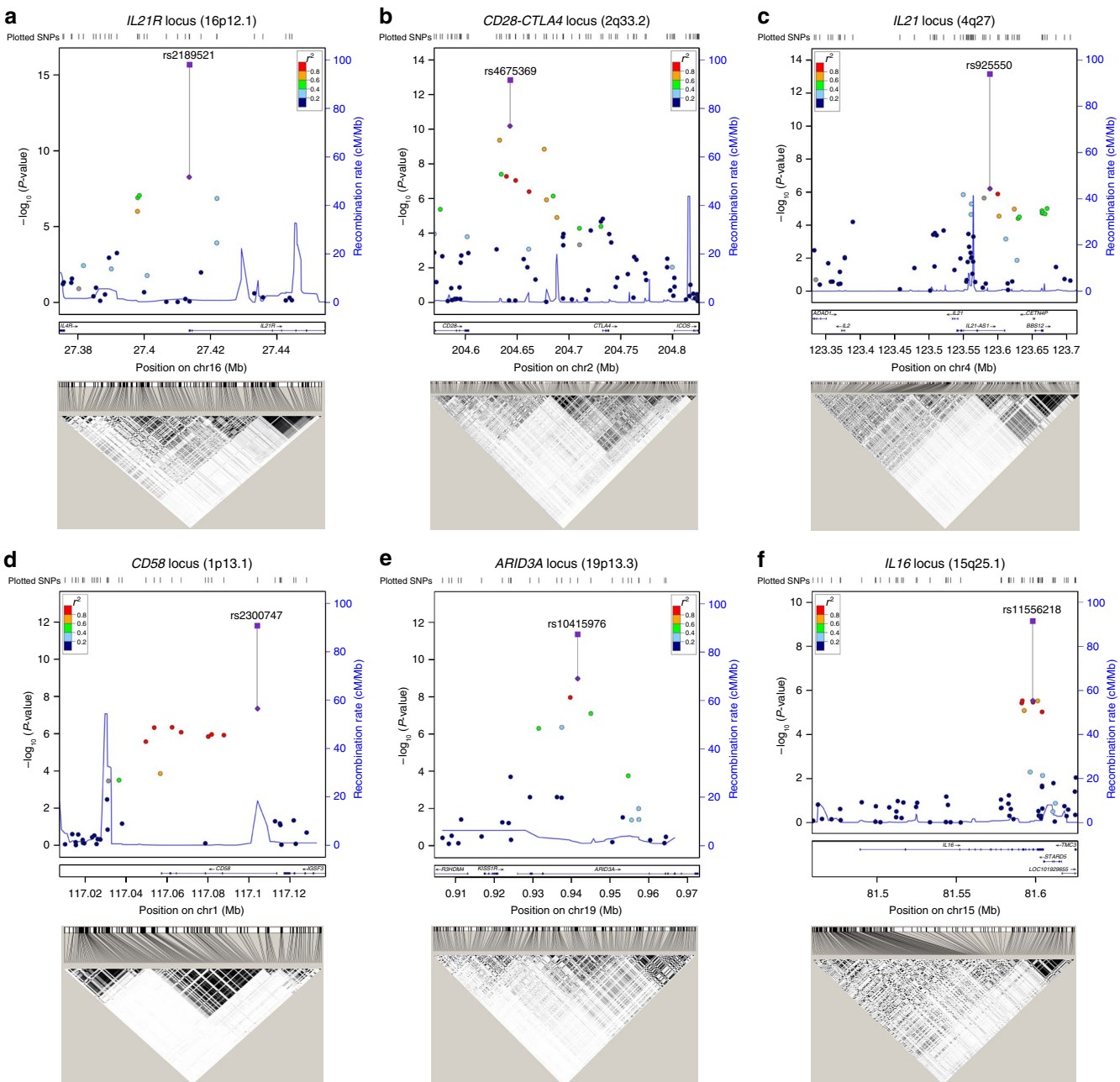

**Figure 2 | Regional association plots of the six novel loci associated with PBC.** The SNP chromosomal location on genome build hg19 is indicated on the x axis and the −log10 of P-value for each SNP is plotted on the left-hand y axis. Genes and ESTs within the region are shown in the lower panels. The Locus Zoom[36] plots of GWAS P-value and LD ($r^2$) of SNPs with the most significant SNP are shown by the colour codes, depending on their expected degree of correlation ($r^2$) with the top SNP (as estimated internally by LocusZoom on the basis of 1000 Genomes Asian haplotypes from March 2012). The square dots with line to the most significant SNP indicate the combined P-value with the discovery and replication panels. Shown below each Locus Zoom plot is the $r^2$-based LD map that is based on the genotype data of 208 Han Chinese (CHB + CHS) 1000-genome-project samples, using Haploview 4.1 program[34]. (**a–f**) Chromosome loci 16p12.1 (*IL21R*), 2q33.2 (*CD28-CTLA4*), 4q27 (*IL21*), 1p13.1 (*CD58*), 19p13.3 (*ARID3A*) and 15q25.1 (*IL16*).

that variation in *IL21R* expression may explain this association (Supplementary Tables 10 and 11). We also note that in a recent study of European PBC the genomic region containing *IL21R* and *IL4R* while not achieving significant association did show suggestive association ($P = 1.6 \times 10^{-7}$) (ref. 5).

Multiple SNPs across the *CD28-CTLA4-ICOS* locus reaching genome-wide significance were found at the discovery stage (Supplementary Table 5) and further confirmed by the replication study. On the basis of the functional role of all the three genes in T-cell costimulatory pathways, the *CD28-CTLA4-ICOS* locus has long been suggested by many candidate gene studies as a PBC risk

locus, but large-scale GWAS and dense fine-mapping studies in European PBC cohorts have not revealed this association[2–7]. An earlier study in a small Chinese PBC cohort also suggested association of the *CTLA4* variant rs231775 with PBC[15]. Our GWAS study is the first to report the genome-wide significance of the *CD28-CTLA4-ICOS* locus with PBC. The most significant signal was obtained for the intergenic variant rs4675369. Haplotype analysis using the Han Chinese genome sequencing data identified a major recombination breakpoint between *CD28* and *CTLA4*. In addition, rs4675369, together with five other highly linked SNPs ($r^2 > 0.9$), was located in the LD block

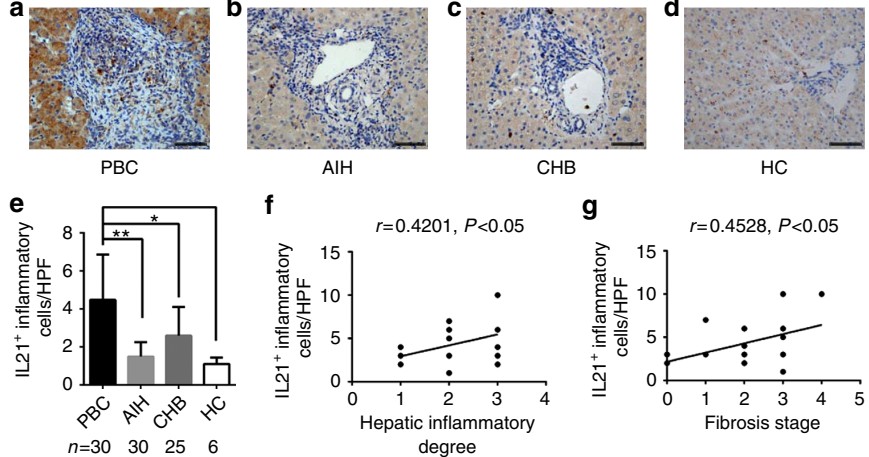

**Figure 3 | Liver immunohistochemical staining of IL21.** (**a–d**) Representative staining images from patients with PBC, AIH, CHB and HC are shown ( × 400). (**e**) Quantification of hepatic IL21 in PBC ($n = 30$), AIH ($n = 30$), CHB ($n = 25$) and HC ($n = 6$), mean ± s.e.m. The frequency of hepatic IL21$^+$ cells is positively correlated with hepatic inflammation degrees (**f**) and fibrosis stages (**g**) in PBC (*$P < 0.05$, **$P < 0.01$). Scale bars, 20 μm.

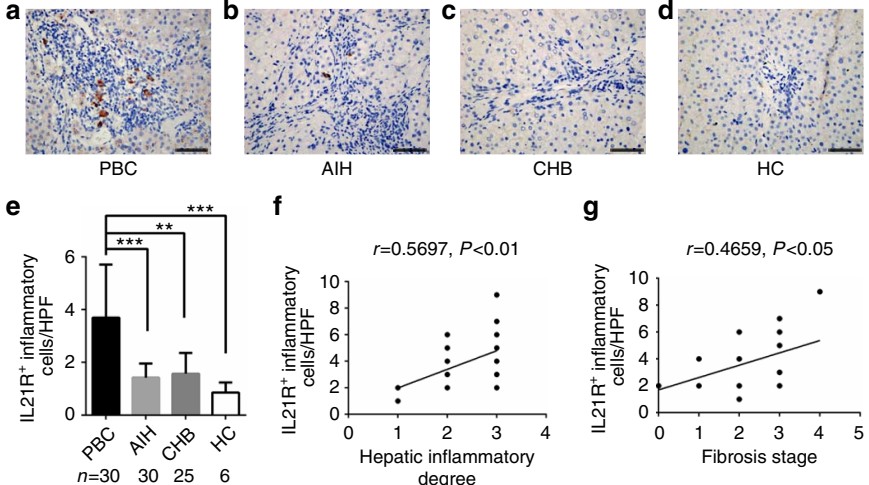

**Figure 4 | Liver immune-histochemical staining of IL21R.** (**a–d**) Representative staining images from patients with PBC, AIH, CHB and HC are shown ( × 400). (**e**) Quantification of hepatic IL21R$^+$ cells in PBC ($n = 30$), AIH ($n = 30$), CHB ($n = 25$) and HC ($n = 6$), mean ± s.e.m. The frequency of hepatic IL21R$^+$ cells is positively correlated with hepatic inflammation degrees (**f**) and fibrosis stages (**g**) in PBC (**$P < 0.01$, ***$P < 0.001$). Scale bars, 20 μm.

containing *CD28*, indicating that the observed significance may be associated with *CD28*, rather than *CTLA4* or *ICOS* (Fig. 2b). Association of the *CD28-CTLA4-ICOS* locus with primary sclerosing cholangitis (PSC) was also reported previously in a Immunochip analysis[16]. However, the variant (rs7426056) associated with PSC is not in LD with any significant variants identified in our study

We identified significant association of the *CD58* locus with PBC. The SNP showing the strongest association, rs2300747 has been previously associated with multiple sclerosis (MS) and rheumatoid arthritis (RA) in large-scale GWAS and replication studies[17,18]. The 1000 genome sequencing data from the Han Chinese population indicated two major haplotype blocks in the *CD58* gene. The first haplotype block includes part of *CD58* intron-1, exon-1 and the 5′ regulatory region and the nearby *IGSF3* gene. The second haplotype block includes most of intron-1 and the remaining *CD58* region (Fig. 2d). The variant

rs2300747 and multiple SNPs in high LD ($r^2 > 0.9$) in the block are eQTLs in lymphoblastoid cell lines (LCLs) (Supplementary Tables 8 and 9). A previous study found that the *A* allele of rs2300747 was associated with lower expression of *CD58* (ref. 19). In our PBC cohort, the *A* allele of rs2300747 is the risk allele, suggesting that decreased expression of *CD58* levels may be related to PBC susceptibility.

Another significant association found in our study, rs10415976, is located in the intronic region of *ARID3A*, a B-cell-restricted transcription factor that activates immunoglobulin heavy chain gene transcription. During the discovery stage of GWAS, five SNPs at the *ARID3A* locus showed significant association, two of them reaching genome-wide significance. These two SNPs were further confirmed in the replication study. All of the significant SNPs, including SNPs in LD ($r^2 > 0.8$), were located in the intronic region and identified as *cis* eQTL loci (Supplementary Tables 10 and 11). ARID3A is the first B-cell-restricted

transcription factor demonstrated to induce autoimmunity[20]. Transgenic mice constitutively expressing ARID3A in B lineage cells produce anti-nuclear antibodies. Systemic lupus erythematosus (SLE) patients show dramatically increased numbers of ARID3A$^+$ B cells compared to healthy controls[21]. Our study is the first to report that the ARID3A locus is associated with human autoimmune of inflammatory disease. Further studies should be aimed at investigating its functional role in the pathogenesis of PBC.

In our GWAS discovery stage, seven SNPs in the IL16 region showed strong association ($P = 9.42$–$2.93 \times 10^{-6}$) with PBC. One of the variants, rs4778636, was selected in the replication study and reached genome-wide significance ($P = 8.99 \times 10^{-9}$). IL16 is a T-cell chemo-attractant factor and has been linked to diseases that are characterized by accumulation of CD4$^+$ cells at the site of inflammation, for example, asthma, atopic dermatitis, RA, MS and Crohn's disease[22]. However, the pathophysiological role of IL16 in PBC remains unclear. IL16 has been found to be constitutively expressed at the mRNA and protein level in monocytes, T lymphocytes, mast cells, eosinophils and dendritic cells. Several of the associated IL16 variants are cis eQTL in peripheral monocytes (Supplementary Tables 10 and 11). This is the first report that IL16 is a disease susceptibility locus with genome-wide significance.

There was also a novel locus CSNK2A2/CCDC113 that did not meet significance in the replication study after correction for multiple comparison, but was significant by genome-wide criteria ($P = 1.51 \times 10^{-8}$) in the combined analysis (Table 2).

**Aberrant expression of IL21/IL21R in PBC liver lesion.** To further explore whether deregulated IL21 and IL21R contribute to the pathogenesis of PBC, we performed histochemical analysis with liver biopsy samples. Significantly elevated levels of expression of IL21 and IL21R was observed in the livers of PBC patients, compared with those of chronic hepatitis B (CHB) patients, autoimmune liver hepatitis (AIH) patients or healthy controls (HC). IL21 was extensively expressed by inflamed hepatocytes and infiltrating inflammatory cells around the portal tracts in PBC patients (Fig. 3a–d), while IL21R$^+$ cells were generally aggregated in the inflamed portal tracts, especially around the damaged interlobular bile ducts in PBC (Fig. 4a–d). Notably, the numbers of IL21$^+$ cells and IL21R$^+$ cells were positively correlated with inflammation severity ($r = 0.45$, $P < 0.05$; $r = 0.43$, $P < 0.05$; respectively) and hepatic fibrosis stages ($r = 0.66$, $P < 0.01$; $r = 0.60$, $P < 0.01$; respectively) in PBC (Figs 3e–g and 4e–g). Therefore, there is marked activation of IL21/IL21R signalling in PBC patients, suggesting that IL21/IL21R interaction is an important component in the pathogenesis of PBC.

## Discussion

Our PBC GWAS included 2,029 cases and 6,163 controls in a Han Chinese population and allowed us to perform comparisons with GWAS findings from European populations (Supplementary Table 12). When three PBC GWAS in European ancestry cohorts from North America and Europe were published, they were hailed by researchers as not only a successful demonstration of the power of GWAS in deciphering complex diseases, but also a landmark achievement in our understanding of the pathogenesis of PBC, given the striking consistency in results among the three studies[23]. This notion was further reinforced later by two separate fine-mapping studies providing virtually identical results with PBC cohorts from Europe and North America[6,7]. These findings prompted some experts to proclaim that PBC probably does constitute a single disease entity across different populations[23]. However, a subsequent GWAS from a Japanese PBC cohort with

divergent results, failed to identify any non-MHC loci with genome-wide significance during the discovery stage. The most significant non-MHC locus reaching genome-wide significance after combined analysis is TNFSF15, which is not associated in European PBC cohorts. Only three loci identified in the Caucasian cohorts were replicated in Japanese cohort after combined analysis[8]. Lack of a large sample size, as well as ethnic specific chips, may have contributed to these results[24]. Compared with the European and Japanese cohorts, the Han Chinese PBC cohort offers an advantage in genetic studies, since the overall genetic variation in the Han Chinese population is not as great as observed in the European and Japanese populations[25,26]. The low $\chi^2$ inflation prior to PCA in our study ($\lambda_{gc} = 1.029$) further supports these previous observations. The low genetic variation may enhance the power of association studies in our Han Chinese PBC cohort because of reduced genetic heterogeneity.

GWAS analysis has achieved great success in our understanding of PBC pathogenesis. Multiple PBC risk loci identified in this study were also shared risk loci in other autoimmune diseases (Supplementary Table 13). But several unique features were observed in our studies. Our IL21 results in particular have important implications and this gene might be worth exploring as a therapeutic target. IL21 is predominantly produced by Tfh cells although high levels are also noted in Th$_{17}$ and natural killer (NK) cells. Tfh are characterized by expression of the cell surface marker CXCR5, encoded by the gene located at the PBC loci previously identified and strongly suggested by the current study[4]. Binding of IL21 with its ligand IL21R leads to activation of multiple downstream signalling molecules, including STAT1 and STAT3, which is important for proliferation and differentiation of T cells, B cells and NK cells. We also note that IL12 (both IL12A and IL12RB2 are associated with PBC) has been shown to enhances IL21 production and the frequency of IL21$^+$ CD4 T cells[27].

Two genes (CD80 and TNFSF15) encoding T-cell costimulatory molecules had been previously identified as PBC risk loci and were confirmed in our study. Identification of two more genetic loci encoding CD28/CTLA4/ICOS and CD58 provided further evidence that deregulated costimulatory signalling confers predisposition to the disease.

Association of IL16 locus with PBC may have significant application in PBC treatment. A previous animal study found that neutralizing anti-IL16 mAb can prevent insulitis and type 1 diabetes in mice[28]. Neutralizing IL16 in PBC patients may represent a potential therapeutic approach.

In summary, this is the first report of a large scale GWAS of PBC in a Han Chinese population and has identified six new PBC susceptibility loci. Our study supports the hypothesis that the IL21 signalling pathway and Tfh cells are involved in PBC pathogenesis. Our results also draw attention to the importance of T-cell costimulatory signalling in PBC and advance our understanding of PBC as a disease of immune deregulation.

## Methods

**Patient and control selection and diagnosis.** PBC patients were recruited with the approval of the research ethics boards of Southeast University and Jiaotong University and in accordance with the guidelines of the Declaration of Helsinki (2008). Informed consent was obtained from all subjects recruited. PBC diagnosis in this study was based on the criteria recommended by AASLD and the European Association for the Study of the Liver (EASL)[29,30]. PBC patients were recruited from two main sources, the Jiangsu Province PBC Collaboration Group (JSPPCG) and Renji Hospital of Jiaotong University. Patient blood samples from Renji Hospital were tested for PBC-specific autoantibodies with EUROLINE Profile Autoimmune Liver Diseases (DL1300-1601-5G). JSPPCG samples were collected from participating hospitals and PBC-specific autoantibodies were tested by different clinical laboratories. Samples from the JSPPCG collection were verified for autoantibody status with an in-house ELISA assay for anti-PDC-E2 and a 3E assay

with a commercial ELISA kit (Shanghai Kexin Biotech, Shanghai, China). We confirm our study is compliant with the 'Guidance of the Ministry of Science and Technology (MOST) for the Review and Approval of Human Genetic Resources', which requires formal approval for the export of human genetic material or data from China.

The control samples used in the discovery stage were recruited from previous studies[31,32]. Healthy control samples used in replication stage were recruited in Nanjing, from the annual health check of staff members of Southeast University. All the control samples used in the replication study have been tested for AMA-M2 autoantibody using 3E-based ELISA assay. Only AMA-M2 negative samples were retained for study. To avoid potential population stratification, the vast majority of PBC patients and controls for GWA and replication studies originally came from Central China (Supplementary Fig. 1). All samples from both the discovery and replication stages were unrelated individuals of self-claimed Chinese Han descent. The demographic information of patients and controls were summarized in Supplementary Tables 1 and 2.

**Quality control in the GWA discovery stage.** The Illumina HumanOmniZhongHua-8 Beadchip (v1.1), a population-specific beadchip targeting common, intermediate, and rare variations found within Chinese populations, was used for both PBC and control samples. The Illumina beadchip scan of 1,127 PBC samples was performed by Genenenergy Bio-Technology, an Illumina CSPro-certified service provider. Scanning of 4,074 control samples was conducted at the Key Laboratory of Dermatology at Anhui Medical University (Ministry of Education). PBC and control samples with overall call rates of <98% were excluded from further analysis. Unexpected duplicates or probable relatives were excluded based on pairwise identity-by-state comparisons using the 'PI_HAT' value in PLINK (all PI_HAT>0.25)[33]. The remaining samples were subsequently assessed for population outlier and stratification using a PCA-based approach. PCA was performed using the smartpca software[33]. After filtering, 1,122 cases and 4,036 controls were retained for analysis. In the GWA stage, single-marker association analyses were performed using logistic regression with gender as a covariate. The Manhattan plot of $-\log10 P$ was generated using Haploview v4.2 (ref. 34). The Quantile–quantile (Q–Q) plot was used to assess the number and magnitude of observed associations between genotyped SNPs and PBC, compared with the association statistics expected under the null hypothesis of no association. The Q–Q plot was created using the R qq.plot function.

We performed systematic quality control on the raw genotyping data to filter out both unqualified samples and SNPs. After excluding mitochondrial SNPs and SNPs on sex chromosomes, the following criteria were applied to disqualify additional SNPs: SNPs with a call rate <98%, or with minor allele frequency (MAF)<0.01 in all samples or SNPs with genotype distributions that deviated from those expected by the Hardy–Weinberg equilibrium ($P<1\times10^{-4}$) in the controls. For all the SNPs with $P<1\times10^{-5}$ in the discovery stage, manual checking of SNP genotyping clustering in each dataset was performed and 54 SNPs were further excluded for the analysis. After quality control filtering, there were 776,516 SNPs remained in the discovery stage (Supplementary Table 3).

**SNP selection and genotyping in the replication stage.** SNPs for the replication stage were selected using the following criteria: each locus with one SNP reaching $P<1\times10^{-6}$ or with minimum three SNPs reaching $P<5.0\times10^{-6}$ for discovery samples. A total of 22 non-MHC loci matched these criteria, except *TNFRSF1A* locus, which has three SNPs reaching $P<6\times10^{-5}$. The most significant SNPs in *HLA-DRA* locus (rs9268644) and *DPB1* locus (rs9501251) were selected for validation. For all the known PBC loci published previously, one of the most significant SNPs was selected for validation. We also intentionally selected one SNP (rs1944918) in *POU2AF1* locus, since this locus was reported in Japanese GWAS as the second most significant locus for PBC. For each loci newly discovered, two of most significant SNPs were selected for validation. Genotyping analyses of replication were conducted by the Sequenom MassARRAY system using the iPlex method[35]. All SNPs achieved a call rate >98% and had no deviation from HWE ($P<0.05$) in the controls.

**Statistical analysis in the replication and combined stage.** For the replication studies, 34 SNPs that passed quality control were analysed using logistic regression with gender as a covariate, assuming an additive allelic effect. The P-values adjusted by gender were reported without correction for multiple testing. Joint analysis of all combined samples of Han Chinese origin was conducted by both the random effects model ($I^2>25\%$) and by the fixed-effect model ($I^2<25\%$). The P-value reported in Tables 1 and 2; Supplementary Table 6 was based on the results by the fixed-effect model. The regional association plot was created using LocusZoom[36]. Imputation of selected loci was conducted using IMPUTE2 software version 2.2.2 and version 3 of 1000 Genomes Project data as the reference set[37].

We used seeQTL (http://www.bios.unc.edu/research/genomic_software/seeQTL/) and the University of Chicago eQTL browser (http://eqtl.uchicago.edu/cgi-bin/gbrowse/eqtl/) to identify eQTLs amongst significant variants[38–40]. eQTLs that are in high linkage disequilibrium ($r^2>0.8$) with the most strongly associated

SNP at the locus were extracted. Gene regulatory elements from the Encyclopedia of DNA Elements (ENCODE) database were annotated using HaploReg[41]. For each locus, SNPs in high linkage disequilibrium ($r^2>0.8$) with the most significantly associated SNP was assessed as to whether they lie within regions with promoter and enhancer histone marks, DNase-I hypersensitivity, protein binding or regulatory motifs in one or more of 147 cell types.

**Immunohistochemistry.** Paraffin-embedded liver tissues, derived from ultrasound-guided needle liver biopsies of 30 PBC patients, 30 AIH patients, 25 CHB patients and 6 controls, were studied. Sections were stained with either hematoxylin and eosin (H&E) or Masson and were independently reviewed 'blindly'. Inflammatory degrees and fibrotic stages were graded according to the Scheuer scoring system. The six healthy liver tissues were collected from donors whose livers were subsequently used for liver transplantation. Immunohistochemistry imaging of IL21 and IL21R staining of human liver tissues was performed on a Leica Bond system (Leica, Germany) using the standard protocol. The liver sections were pre-treated using heat-mediated antigen retrieval with sodium citrate buffer (pH 6, epitope retrieval solution 1) for 20 min followed by incubation with an anti-IL21 antibody (1:200 dilution, ab5978, Abcam, Cambridge, MA, USA) or an anti-IL21R antibody (1:150 dilution, ab13268, Abcam) for 20 min at room temperature and then detected using a horse radish peroxidase-conjugated compact polymer system. 3,3′-diaminobenzidinewas used as the chromogen[42]. The liver sections were then counterstained with hematoxylin. All the sections were visualized using light microscopy (Olympus, Japan), and five fields were randomly selected for each section. The numbers of IL21- or IL21R-positive cells were quantified at $40\times10$ magnification. All the samples were analysed by a hepatic pathologist. All data are reported as mean ± standard error (s.e.). The Mann–Whitney $U$-test was used to evaluate differences in continuous variables. A P-value of <0.05 was considered statistically significant. Correlations were determined using the Spearman's correlation coefficient. All analyses were two-tailed and performed using Prism software Version 6.0 (Graphpad Software, La Jolla, CA, USA).

**Data availability.** The GWA SNP results are available upon request by contacting X.L. at xiangdongliu@seu.edu.cn. Any additional data (beyond those included in the main text and Supplementary Information) that support the findings of this study are also available from the corresponding author upon request.

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

## Acknowledgements

We thank all participating members of The Jiangsu Provincial PBC Collaboration Group for providing patient samples and clinical information. We would like to thank the patients for their participation in this study. This work was supported in part by grants from the National Natural Science Foundation of China (No. 81270295 to Xiangdong L.; 81325002 and 81620108002 to X.M.; No. 81272737 to W.C.; No. 81400608 and 81570469 to R.T.; Nos. 81130058, 81430034 and 91542123 to Z.L.; and No. 31301113 to X.S.), by grants from Jiangsu provincial Healthy fund (No. BL201406 to W.C.), by grants from Jiangsu Natural Science Foundation (SBK2015020697 to Xiangdong L. and BK20130607 to X.S.) and by grant from the Innovation Program of Shanghai Jiao Tong University (YG2014MS43 to R.T.). This work was also supported by NIH Grant R01DK091823 to M.F.S. and M.E.G. Xiangdong L. is a fellow at the Collaborative Innovation Center for Cardiovascular Disease Translational Medicine and the recipient of the Jiangsu Provincial Innovation fund.

## Author contributions

This study was initially conceived and designed by Xiangdong L., X.M., W.C., Z.L., X. Zuo; the collection of samples and clinical information for the study were performed by F.Q., R.T., X.S., Y.D., L.W., P.X., X. Zhu, Jian W., Jianfang W., Y. Gong, C.H., Y. Gao, K.Z., Y.J., J.Z., Y.S., Z.H., Y.T., H.Z., N.D., L. Liu, X.W., W.Z., Z.Z., J.N., W.S., Y.Z., Y.M., X. Zha., P.J., H.J., Q.D., Jun. L., Z.L., X.B., L. Li, M.L., Q.W., Y.Y., F.Y., M.L., X.J., X.X., Y.L., J.F., D.Q., Z.Z., H.Q., S.C., B.J., P.L., G.C., T.W., Y.S., J.Y., H.T., M.H., M.X., H.P., L.H., Z.Q., J.G., L.J., W.T., Xiangdong L.; Processing of samples for the study and experimental work were supervised by Xiangdong L., X.S., X.M., R.T., J.H., W.X., J.Z. and performed by X.S., H.Z., M.D., Jin.L., P.Z., C.W., Y.Z., P.J., Y.W., R.J., J.X., Y.Z.; histochemical analysis was performed by Y.W.; the statistical analyses of the data were performed by X. Zuo, X. Zhe,, Q.H., G.H., Z.S., Xiangdong L.; the paper was written by Xiangdong L. and critically reviewed and revised by M.F.S., Xiang L., M.E.G.

## Additional information

**Competing interests:** The authors declare no competing financial interests.

