## [Peer Review File · Nature Communications]

Reviewers' Comments:

Reviewer #1 (Remarks to the Author)

Important study which highlights genetic risk for PBC in a Han Chinese population. Clear English.

- 1) the abstract should be a little clearer; pathogenesis of PBC is clearly more than just autoimmunity, yet Genetic studies that are cross-sectional like this, provide data only on risk of developing the diagnostic features of disease (in this case ALP/AMA) as opposed to the clinically diverse outcomes
- 2) No HLA data is presented- perhaps the authors want to publish separately but from a reader's perspective they want as much useful data as possible;
- 3) Very unclear as to how SNPs for validation were actually chosen, including number of SNPs meeting criteria: can we have a table that specifies very clearly how and why SNPs were chosen, and the definitions used for replication, GWAS significance. Could give QC data.
- 4) A table comparing these findings with the Japanese and Caucasian studies would be very helpful.
- 5) Did the authors want to do any meta-analysis of loci based on published gene frequencies and P values from the Japanese and international meta-analyses that are published?
- 6) Did I miss the patient demographics?
- 7) The discussion is long and speculative
- 8) Is the CD28 association the same as for PSC? CTLA-4/CD28 locus is complex to analyse.
- 9) What is the overlap with other autoimmune diseases- will a table help here as well?

Reviewer #2 (Remarks to the Author)

Dear authors,

Thank you for studying PBC in a Chinese population. There is for sure a need to study the genetics factors in multiple ethnicities. It is assuring that you find some overlap and very exciting that you find novel findings not previously reported. The work is very well done,

The reviewer recognize that a GWAS is a major undertaking with a considerable degree of team-work. Nevertheless, the inclusion of >20 sharing first-authors is way out of what is normally done in these kind of papers. I believe that the authors should agree on who actually did the most work and give them due credit with a maximum of 3-4 first authors. The same goes with the corresponding authors, in this paper there is 4 corresponding authors, who should the reader contact if he/she has any questions? This should not be more than 2.

Major comments

The Q-Q plot both with and without the HLA shows a remarkable deviation from the Null. Is this due to the uniform ethnicity? In European populations such a large deviation is not normally seen.

Regarding the MHC region why was only two SNPs from the MHC region chosen? I believe that there would be many independent signals here as normally seen.

The overlap of the findings with other autoimmune diseases both in Chinese and Caucasians are mentioned throughout the manuscript. Can the authors systematically present the overlap with previous findings in a table?

The authors point out that IL21R also showed suggestive association in European patients. It

would be interesting to see if the immunohistochemical stainings are the same in Europeans. Both an overlap of findings and a lack of correlation can be the case and will tell us if the pathogenesis is actually the same.

Minor comments:

The sequencing data for TNFSF15 do not need to take up a paragraph. Just briefly mention and refer to the table.

Could the differences in the IL12A in Caucasians vs. Chinese be illustrated with a figure ?

NCOMMS-16-23721-T

Reply to reviewers' comments

Reviewer #1 (Remarks to the Author):

Important study which highlights genetic risk for PBC in a Han Chinese population. Clear English.

Thanks!

1) the abstract should be a little clearer; pathogenesis of PBC is clearly more than just autoimmunity, yet Genetic studies that are cross-sectional like this, provide data only on risk of developing the diagnostic features of disease (in this case ALP/AMA) as opposed to the clinically diverse outcomes

Reply: We modified abstract accordingly

2) No HLA data is presented- perhaps the authors want to publish separately but from a reader's perspective they want as much useful data as possible;

Reply: We agree that the MHC associations with any autoimmune disease is important. However, it is also complex and ideally would include imputation and analyses of all the various HLA antigens and amino acids as well as conditioning studies to determine independent signals. We are indeed in progress of performing large scale imputation analysis with deep sequencing data of 20,635 Han Chinese samples from 5Mb of MHC region. We believe this is more suitable as a separate manuscript. The current study focuses on the non-MHC associations.

3) Very unclear as to how SNPs for validation were actually chosen, including number of SNPs meeting criteria: can we have a table that specifies very clearly how and why SNPs were chosen, and the definitions used for replication, GWAS significance. Could give QC data.

Reply: We generated supplementary table 3 for QC data and described more clearly how SNPs were selected for validation in method section "SNP Selection and Genotyping in the Replication Stage".

4) A table comparing these findings with the Japanese and Caucasian studies would be very helpful.

Reply: The table comparing our findings with the Japanese and Caucasian studies has been generated as "Supplementary Table 12".

5) Did the authors want to do any meta-analysis of loci based on published gene frequencies and P values from the Japanese and international meta-analyses that are published?

Reply: A meta-analysis combining our data with Japanese and Caucasian data is being conducted by the international collaboration group.

6) Did I miss the patient demographics?

Reply: The table containing patients' demographics has been generated in "Supplementary Table 3".

7) The discussion is long and speculative

Reply: The discussion has been rewritten accordingly.

8) Is the CD28 association the same as for PSC? CTLA-4/CD28 locus is complex to analyse.

Reply: Association of CD28-CTLA4-ICOS locus with primary sclerosing cholangitis (PSC) was reported previously in a ImmunoChip analysis. The variant (rs7426056) associated with PSC is not in LD with any significant variants identified in our study. There is no GWAS report for PSC in Han or Asian populations and no report for candidate gene study in this region with Han or Asian PSC cohorts. Therefore, we do not have sufficient information to make any conclusions regarding the potential overlap of a variant(s) at this locus for these two diseases.

9) What is the overlap with other autoimmune diseases- will a table help here as well?

Reply: A table to present these overlap loci has been generated as "Supplementary table 13".

Reviewer #2 (Remarks to the Author):

Thank you for studying PBC in a Chinese population. There is for sure a need to study the genetics factors in multiple ethnicities. It is assuring that you find some overlap and very exciting that you find novel findings not previously reported. The work is very well done,

Thanks!

The reviewer recognize that a GWAS is a major undertaking with a considerable degree of team-work. Nevertheless, the inclusion of >20 sharing first-authors is way out of what is normally done in these kind

of papers. I believe that the authors should agree on who actually did the most work and give them due credit with a maximum of 3-4 first authors. The same goes with the corresponding authors, in this paper there is 4 corresponding authors, who should the reader contact if he/she has any questions? This should not be more than 2.

Reply: We appreciate the reviewer's concern however we believe that this needs to be balanced with appropriate recognition of contributions. We have now complied with the journal's formal requirement that allows for a maximum of 6 equally-contributing authors and 3 corresponding authors. We informed all the authors for the change of authorship to comply this policy and the manuscript now includes 6 equally-contributing authors and 3 corresponding authors.

Major comments

1. The Q-Q plot both with and without the HLA shows a remarkable deviation from the Null. Is this due to the uniform ethnicity? In European populations such a large deviation is not normally seen.

Reply: We agree that the Q-Q plot shows more deviation from the null than in studies of European populations. As the reviewer suggests it is possible that the increased ethnic uniformity could contribute to higher power (as we have commented elsewhere in the text) and thus more deviation. However, it is also possible that the GWAS panel and Han Chinese participants in this study has resulted in more SNPs in linkage disequilibrium with the causative variation. This is speculative and we have included a brief comment in this regard.

2. Regarding the MHC region why was only two SNPs from the MHC region chosen? I believe that there would be many independent signals here as normally seen.

Reply: As indicated in our response to Reviewer 1, we believe that the MHC deserves strong attention but the current study focuses on the non-MHC associations. The complex MHC associations will be pursued in depth as a separate set of analyses.

3. The overlap of the findings with other autoimmune diseases both in Chinese and Caucasians are mentioned throughout the manuscript. Can the authors systematically present the overlap with previous findings in a table?

Reply: This comment is the same question as Review 1 (comment 9). A table to present these overlap loci has been generated as "Supplementary table 13".

4. The authors point out that IL21R also showed suggestive association in European patients. It would be interesting to see if the immunohistochemical stainings are the same in Europeans. Both an overlap of findings and a lack of correlation can be the case and will tell us if the pathogenesis is actually the same.

Reply: We agree with the comments by the reviewer. But practically it is not possible for us in China to obtain such specimens.

Minor comments:

1. The sequencing data for TNFSF15 do not need to take up a paragraph. Just briefly mention and refer to the table.

Reply: We agree with the reviewers and revised the text.

2. Could the differences in the IL12A in Caucasians vs. Chinese be illustrated with a figure ?

Reply: We generated the “Supplemental Table 8” to present the differences in IL12A locus.

Reviewers' Comments:

Reviewer #1 (Remarks to the Author)

Whilst the authors have not directly addressed all the points I raise, they have made cogent responses, and I have nothing further to add with this nice study.

Reviewer #2 (Remarks to the Author)

Dear authors,

Thank you for clarifying all remaining issues, the paper has improved and I believe it will be an important addition to the literature.